# Pruritogenic Mediators and New Antipruritic Drugs in Atopic Dermatitis

**DOI:** 10.3390/jcm12062091

**Published:** 2023-03-07

**Authors:** Dimitra Koumaki, Stamatios Gregoriou, George Evangelou, Konstantinos Krasagakis

**Affiliations:** 1Dermatology Department, University Hospital of Heraklion, 71110 Heraklion, Greece; 2Department of Dermatology and Venereology, Andreas Sygros Hospital, Medical School of Athens, National and Kapodistrian University of Athens, 16121 Athens, Greece

**Keywords:** atopic dermatitis, pruritogen, pruritus, histamine, drugs

## Abstract

Atopic dermatitis (AD) is a common highly pruritic chronic inflammatory skin disorder affecting 5–20% of children worldwide, while the prevalence in adults varies from 7 to 10%. Patients with AD experience intense pruritus that could lead to sleep disturbance and impaired quality of life. Here, we analyze the pathophysiology of itchiness in AD. We extensively review the histamine-dependent and histamine-independent pruritogens. Several receptors, substance P, secreted molecules, chemokines, and cytokines are involved as mediators in chronic itch. We also, summarize the new emerging antipruritic drugs in atopic dermatitis.

## 1. Introduction

Atopic dermatitis (AD) is a common, chronic, pruritic, inflammatory skin disease that occurs in nearly 20 percent of children worldwide and largely varies between ethnic groups and countries [1,2,3]. There are limited data regarding the prevalence of AD in adulthood. Population-based studies from northern Europe have reported a prevalence rate of AD in adulthood between 10 and 14 percent [4,5]. A cross-sectional study from the United States, which included almost 1300 adults, reported a prevalence of 7.3 percent (95% CI 5.9–8.8) [6]. The prevalence of AD among patients between 60 and 69 years old in Saarland, Germany, was 4% [7]. In Poland, the prevalence of AD was 2% for those over 60 years old, compared to 5% for children (ages 6–7), 4% for teenagers (ages 13–14), and 3% for adults (20–44 years) [8]. Similarly, 4% of adults (30–39 years) and 3% of elderly patients (60 years or older) who attended a hospital in Tokyo had AD [9], compared to 19% of young patients (7–9 years) in Otsu, Japan [10,11]. Risk factors for AD include various genetic and environmental factors [12,13,14,15,16,17,18]. AD is linked to cutaneous hyperreactivity to triggers in the environment that are safe for healthy, non-atopic people [19].

Increased levels of both total serum IgE and IgE that is specific to common allergens are frequently present in these patients. Based on this hyperreactivity, two separate subtypes of AD have recently been identified: an “intrinsic” form of AD that lacks detectable IgE-mediated sensitization and an “extrinsic” form linked to IgE-mediated sensitization to environmental allergens [20]. According to recent studies, the prevalence of extrinsic and intrinsic AD is, respectively, 70–80% and 20–30%. Environmental allergens including animal dander, grasses, and pollens, as well as certain foods like milk, eggs, and wheat, can exacerbate AD [11]. Children are more susceptible to food allergy-related symptoms than older patients, while adult and elderly AD patients with moderate-to-severe cases of the disease tend to be more sensitive to food allergens than patients with milder AD [11]. In a study with AD patients, six inhaled allergens were most frequently identified in patients with high allergen-specific IgE (44 or 73.3%), with the majority of allergens coming from timothy grass pollen (43.2%). The second allergen most frequently identified (25.0%) was birch pollen. Extremely high levels of total IgE, thymus and activation-regulated chemokine levels, and peripheral blood eosinophil counts were present in elderly patients with AD [21]. Cross-reactions may be the cause of the high incidence of these inhaled allergens, as both timothy grass and birch grass pollen have many elements in common with other allergens. Birch pollen frequently reacts adversely with allergens in fruits, nuts, and vegetables [22]. The levels of total IgE, but the tryptase levels, have been associated with the severity of the symptoms in atopic dermatitis patients [23]. More effective candidate biomarkers for determining the severity of AD may be serum LDH and IL-18 [24]. Older AD patients are less allergic to foods, animal dander, and fungi, and more sensitive to dust mites and pollens, as indicated by elevated serum IgE levels that are specific to these allergens [25]. The relationship between serum IgE and treatment response has been controversial. Some studies have shown that dynamic changes in serum IgE levels may reflect the way AD responds to treatment, while others have shown that there is only a weak relationship between serum IgE and continued disease severity after treatment, suggesting that serum IgE is not the most accurate biomarker of AD [25].

Chronic pruritus is a defined symptom of AD that varies from 87 to 100 percent, but in reality, all AD patients who have the active disease also experience pruritus [26]. In addition to pruritus, AD patients experience skin pain [27]. Uncontrolled pruritus in AD results in impaired quality of life, loss of work productivity, and sleep disturbance [28,29]. Although the pathophysiology of AD’s pruritus still needs to be elucidated, it is widely accepted that the signals’ transmission through unmyelinated, histamine-sensitive, and non-histamine-sensitive peripheral C-nerve fibers plays a crucial role [30,31,32]. Here, we summarize existing knowledge on how keratinocytes, the immune system, and the nervous system contribute to the pathophysiology of chronic itch in AD and discuss possible therapeutic strategies to target these circuits.

## 2. Pruritogenic Mediators

One of the distinguishing characteristics of atopic dermatitis (AD) is chronic pruritus. Between 87 and 100% of AD patients have chronic pruritus at any given time [33,34]. Studies have distinguished histamine-sensitive and histamine-insensitive pruriceptive sensory nerves in the cutaneous neuronal network (Figure 1, Table 1).

### 2.1. Histamine-Dependent Pruritogens

Mast cells are the main source of histamine production, but platelets, basophils, and neutrophils are all involved. Histamine receptors H1, H2, H3, and H4 act as a conduit for histamine’s actions.

In the cutaneous neuronal network, studies have differentiated the pruriceptive sensory nerves that are histamine-sensitive and histamine-insensitive [26,28]. Antihistamine medications have shown either slight or no effect on pruritus on AD. This finding suggests that, at least through the activation of H1 receptors, histamine has only a modest role in the pruritus associated with AD [26,28]. Nevertheless, histamines may still be involved in AD-related inflammation and pruritus. In an experimentally-induced pruritus study, the histamine H(4) receptor antagonists’ inhibitory effect was superior to those examined with histamine H(1) receptor antagonists [36]. However, clinical trials reported no noteworthy reduction in pruritus in patients with AD who received a selective H_4_ receptor antagonist [37,38].

Significantly, it has now been demonstrated that pruritogen receptors for substances generated from immune cells, such as leukotriene C4 (Cysltr2), histamine (Hrh1), IL-4 (Il4ra), IL-13 (Il13ra1), and IL-31 (Il31ra and OSMR), are expressed on the same neuronal subsets [39].

### 2.2. Platelet Activating Factor (PAF)

Phospholipase A2 acts on human mast cells to induce the production and secretion of platelet-activating factor (PAF). Platelets, neutrophils, monocytes, macrophages, and endothelial cells all continually produce platelet-activating factor (PAF), a phospholipid, in a very low quantity [40,41]. When there is inflammation, an allergic reaction, activation of thrombotic cascades, blood vessel dilation, platelet aggregation, or shock, its synthesis is elevated. The PAF and PAF-like phospholipids are destroyed by PAF-acetyl hydrolase (PAF-AH), which regulates their activity [42]. Preliminary evidence indicates that PAF may also be involved in human anaphylaxis, especially in the amplification of the initial response. As an example of a second-generation antihistamine agonist, rupatadine impacts the PAF and histamine pathways, which further prevents mast cell activation. The use of this dual braking is permitted in the management of chronic urticaria and allergic rhinitis [42,43,44,45]. When PAF is intradermally injected, then wheal and pruritus are evoked. This is due to histamine release by mast cells (MCs) through a neurogenic pathway as it can be blocked by the nerves [45,46].

In the study conducted by Gomułka et al. [40], circulating serum eosinophil-derived neurotoxin EDN, PAF, and vascular endothelial growth factor (VEGF) were not significantly correlated with the severity of pruritus in patients with AD. Instead, they reported that patients with AD had significantly higher serum levels of VEGF and EDN compared to the control group [40]. A study from Japan reported an improvement in total pruritus score in AD patients after administration of an H1 antihistamine, rupatadine, together with a PAF antagonist [47].

## 3. Histamine-Independent Pruritogens

### 3.1. Protease and (PARs) Protease-Activated Receptors

The homeostasis of the skin barrier depends on proteases. Abnormal protease activity contributes to the breakdown of the epidermal barrier, which then triggers the release of inflammatory cytokines by activating the associated receptors [48,49]. PARs operate as a mediator for the diverse activities of proteases. There are four PARs, or G protein-coupled receptors: PAR1, PAR2, PAR3, and PAR4. They can be activated by proteases produced both internally by immune cells and keratinocytes as well as externally by fungi, *S. aureus*, and dust mites, all of which are crucial AD triggers [38]. The functional role of PAR2 has been described in AD more clearly than those of the other three receptors. In the lesioned skin of AD patients, protease activity and PAR2 expression in keratinocytes and PAR+ peripheral nerves were noticeably elevated [50,51]. Specific serine and cysteine proteases such as tryptase, kallikreins (KLK), and cathepsin can activate PAR2. It has been established that KLK5 activates PAR2 to cause TSLP overexpression in keratinocytes via nuclear factor B [52]. Nevertheless, subsequent studies have demonstrated that KLK5 provoked changes in skin architecture similar to those seen in AD in a PAR2-independent way [52,53]. These studies also showed that independent of skin inflammation and PAR2, KLK7 stimulates an itch response. This shows that non-PAR2-mediated reactions have a role in the development of AD as well [54,55].

Two groups have recently shown the regulatory function of neuronal alteration by PAR2 in keratinocytes in a Grhl3PAR/+ mouse model overexpressing PAR2 in suprabasal keratinocytes. Spontaneous AD-like dermatitis and increased scratching behavior were found in these mice, and skin inflammation was exacerbated by house dust mite (HDM) protease, a major source of allergens in AD [56]. A different study, however, demonstrated that an HDM challenge is required for AD-like dermatitis and elevated scratching behavior [57]. The diverse genetic backgrounds, the age of the mice, and the living conditions could all contribute to the phenotypic discrepancies. After HDM exposure, Grhl3PAR/+ mice showed subclinical epidermal barrier damage and increased nerve fiber density. Importantly, in mice overexpressing epidermal PAR2, PAR2 expression in the DRG was increased even in the absence of HDM exposure [56]. Additionally, skin disruption in Grhl3PAR/+ mice treated with HDM increased the expression of genes linked to inflammation and pruritus in sensory neurons, including TSLPR, IL-31 receptor, and brain-derived neurotrophic factor (BDNF) [56,57]. There is evidence that there is PAR 2 neuroepidermal communication in keratinocytes. PAR2 can be found in primary sensory neurons, peripheral nerves, and keratinocytes. Different roles for PAR2 and DRG in keratinocytes may exist [58]. Primary sensory neurons also contain PAR2, in addition to keratinocytes and peripheral nerves. Some proteases have the ability to sensitize TRPV1 channels or directly activate PAR2+ sensory neurons, which may be a factor in non-histaminergic pruritus [59]. For example, it has been demonstrated that a cysteine protease, called cathepsin S, causes mice to scratch by activating PAR2 in TRPV1 sensory neurons [60]. In a mouse model of AD, pruritus and inflammation response were reduced by PZ-235, a PAR2 pepducin [61]. The use of topical E6005, a phosphodiesterase 4 inhibitor that is related to the PAR2 pathway, demonstrated an antipruritic effect [62].

### 3.2. Thymic Stromal Lymphopoietin (TSLP) and TSLP Receptor TSLPR

Thymic stromal lymphopoietin (TSLP) is an epithelial-cell-derived cytokine that is important in initiating allergic inflammation. There are two different isoforms of TSLP: the short- and long-form [63,64]. While the short variant is continuously expressed and is involved in homeostasis, the long form is induced during inflammation [65,66]. Allergens, bacteria, and viruses are suspected of acting as triggers to promote TSLP synthesis in inflamed tissue [67]. TSLP receptor is mainly expressed by the dendritic cells (DCs) and is a heterodimer composed of the IL-7Rα chain and the TSLP-specific receptor [68]. Raised levels of TSLP are expressed in keratinocytes of lesional skin and serum of patients with atopic dermatitis [69]. TSLP plays a major role in Th2 immune deviation via action on the DCs of the skin [70]. Recent studies have shown that TSLP can activate Langerhans cells and DCs to induce T follicular helper cells, which are linked to AD pathogenesis and severity [71,72]. It has been reported that in AD patients, CD4 + T cells express higher amounts of TSLPR [73]. Additionally, TSLP can damage the skin barrier by inhibiting the expression of filaggrin (FLG), a crucial protein for the growth and upkeep of the skin barrier via the STAT3 and ERK pathways [74]. Notably, TSLP produced by keratinocytes can promote “atopic march” (the progression of atopic disorder to the onset of allergy-induced rhinitis and asthma) and aggravate allergy-induced asthma [75].

The cytokine thymic stromal lymphopoietin (TSLP), which mediates signaling between epithelial cells and innate immune cells, is thought to be the primary cause of AD and the atopic march. It has been demonstrated that itch is promoted by direct communication between epithelial cells and cutaneous sensory neurons via TSLP [76].

One of the main mediators of serotonergic itch is the serotonin receptor HTR7, (5-Hydroxytryptamine Receptor 7). The ion channel TRPA1, transient receptor potential ankyrin 1, is encouraged to open by HTR7 activation, which in turn brings on itch-related behaviors. Moreover, HTR7 and TRPA1 are necessary for acute itch brought on by serotonin or a selective serotonin reuptake inhibitor [77].

Atopic dermatitis and other persistent itch problems in humans have long been connected to abnormal serotonin signaling [77].

Furthermore, it has been demonstrated that in atopic dermatitis, a subset of basophils stimulates sensory neurons to cause itch flare-ups that are triggered by allergens [78].

Tezepelumab is a human monoclonal immunoglobulin G2-lambda antibody (AMG 157) that binds TSLP and prevents its interaction with the TSLP receptor complex. Tezepelumab is approved by the FDA for add-on maintenance therapy in patients with severe asthma who are ≥12 years of age [79].

In a phase 2a research study (NCT02525094), 113 individuals with AD were randomly selected to receive 280 mg of subcutaneous tezepelumab or a placebo every two weeks along with class 3 topical corticosteroids (TCS). When compared to placebo with TCS, patients receiving tezepelumab showed marginally significant improvements in pruritus numeric rating scale (NRS) from baseline to week 12 (Table 2) [80].

### 3.3. IL-33

IL-33 belongs to the IL-1 cytokine family, is produced by immune, endothelial, and keratinocytes cells, and is upregulated in patients with atopic dermatitis, activating the innate immune system [104,105]. IL-33 stimulates the expression of IL-4 and IL-13 which consequently enhance the expression of IL-31, a pruritogenic interleukin [105]. Furthermore, ILC2 and IL4 are cytokines that both induce pruritus and are stimulated via basophils through the action of IL33. On the other hand, itch mediators including IL-31, substance P, and brain natriuretic peptide (BNP) can also cause the release of cytokines and chemokines that are associated with pain, inflammation, and barrier dysfunction [106,107,108]. Serum levels of IL-33 were linked with the severity of AD and were considerably elevated in AD patients compared to urticaria and psoriasis patients and healthy controls [109]. Additionally, IL-33 levels were highly linked with excoriation and xerosis scores [109]. Another study found that in patients with AD, the level of lichenification and the intensity of pruritus were both linked with the expression of IL-33 in keratinocytes [110]. In addition, IL-33 induces and maintains the Th2 inflammatory response by enhancing the TSLP-DC-OX40L axis [111]. Pruritus in atopic dermatitis is caused in large part by skin-infiltrating neutrophils. CXCL10, a ligand of the CXCR3 receptor that enhances itch by activating sensory neurons, is only produced by neutrophils, and CXCR3 antagonism reduces chronic itch [112]. In a mouse model of allergic contact dermatitis caused by poison ivy, pruritus and skin inflammation were reduced by neutralized antibodies against ST2 or IL-33. Cutaneous inflammation and scratching were both diminished by ST2 expression knockdown in dorsal root ganglia (DRG) neurons [113]. In another study, chronic itch caused by IL-33/ST2 was mediated by astrocytic JAK2-STAT3 cascades, which stimulated TNF- to sensitize gastrin-releasing peptide GRPR signaling. In a 2,4-dinitrofluorobenzene (DNFB)-induced dry skin and allergic contact dermatitis mouse model, it was shown that scratching could be reduced in ST knockout mice [114]. In a phase 2a clinical trial with a human monoclonal antibody against IL-33, etokimab (ANB020), patients with AD showed significant improvements in 5D itch measures and EASI scores [115]. In a phase 2 randomized clinical trial with astegolimumab, a fully human IgG_2_ IL-33 inhibitor, primary and secondary outcomes did not significantly differ between astegolimab and placebo [116].

### 3.4. IL-4 and IL-13

Along with neuropeptides, the production of the alarmins TSLP, IL-33, and IL-25 eventually stimulates the innate and adaptive immune systems in addition to inducing and perceiving itch [14]. The primary Th2 immune response in AD is started and subsequently propagated by this activation. The production of a number of pro-inflammatory mediators occurs next, either directly by type 2 innate lymphoid cells (ILC2) or Th2 effector lymphocytes, or indirectly by stimulating mast cells, basophils, or eosinophils. A number of these mediators can either directly or indirectly increase AD patients’ itching [117,118]. The pathogenesis of AD and pruritus in AD patients are both significantly influenced by the cytokines IL-4 and IL-13. ILC2 and Th2 cells are the main sources of these cytokines’ production and release. They have a variety of impacts on epidermal and dermal cells, as well as on sensory nerve fibers by activating certain receptors that are members of the IL-4Ra chain [117,118]. The ability of IL-4 and IL-13 to sensitize sensory nerves to itch by reducing the sensitivity thresholds to other pruritogenic stimuli, such as histamine, IL-31, and TSLP, has been demonstrated in both in vitro and in vivo studies in mice [119]. Nevertheless, additional research has demonstrated that IL-4 and IL-13 can both directly activate pruritus in mice, and that applying mixtures of these mediators even increased the induction of itch [120]. Transient receptor potential (TRP) V1 and TRPA1, which are unspecific cation channels, are carried by involved sensory nerve fibers [121]. The activation of TRPV1 and/or TRPA1 causes calcium influx after these nerves have been activated by IL-4, IL-13, or IL-31 via their particular receptors, which ultimately causes the release of action potentials via the sodium channels NaV1.7, NaV1.8, or NaV1.9. In order for pruritus to be induced or sensory nerves to be sensitized by these pruritogens, TRPV1 and TRPA1 should be present [121,122]. Due to the downregulation of crucial skin barrier proteins including filaggrin, involucrin, and loricrin, IL-4 and IL-13 also have an impact on the skin barrier and impair its functionality [117,118,119]. According to a recent study, IL-4 increased IL-31/IL-31 receptor signaling, which is crucial for the pathophysiology and transmission of pruritus in AD [123,124]. A pruritic phenotype was induced in the skin of a mouse by transgenic overexpression of IL-13 [125]. Dupilumab, a fully humanized IL-4 receptor α blocker is licensed for the management of atopic dermatitis (AD), and it has been demonstrated to decrease pruritus and the severity of AD [102,126,127].

A completely human anti-IL-13 monoclonal antibody called tralokinumab has been licensed for use in Europe and the United States to treat individuals with moderate to severe atopic dermatitis who are not satisfactorily managed by topical prescription treatments [93]. Randomized trials have assessed the efficacy of tralokinumab in pruritus in AD alone or in combination with topical corticosteroids. In a phase 2b clinical trial (NCT02347176), 204 adults were randomly assigned 1:1:1:1 to receive subcutaneous tralokinumab at doses of 45, 150, or 300 mg every two weeks for 12 weeks, or a placebo, along with concurrent topical glucocorticoids. Upon taking 45 or 300 mg of tralokinumab, participants showed improvements in pruritus Numerical Rating Scale, NRS, (7-day mean) scores from their baseline to week 12 compared to those taking a placebo [93]. A total of 1596 adult patients with atopic dermatitis who were candidates for systemic therapy participated in two identical, randomized phase 3 clinical trials (ECZTRA 1 and ECZTRA 2) and were given 300 mg of subcutaneous tralokinumab as monotherapy or a placebo every other week for 16 weeks [94]. Tralokinumab showed an noticeable reduction of pruritus and an improvement of several scale of AD severity after 16 weeks of monotherapy treatment and a constant response at week 52 [94]. However, in another clinical trial, ECZTRA 3, tralokinumab was used together with topical corticosteroids (TCS) in order to meet the primary and secondary endpoints [95]. In another study, tralokinumab plus TCS resulted in a larger percentage of patients experiencing a decrease in their worst daily pruritus NRS (weekly average) of 4 from their baseline to week 16 compared to placebo plus TCS (45.5% vs. 35.6%). However, this was not statistically significant [96]. In a retrospective study of 12 patients with severe resistant AD who were treated with tralokinumab, using the itch Numerical Rating Scale, it was shown that the severity of the itching was significantly decreased and there was also a great reduction in EASI score [128]. Both dupilumab and tralokinumab may provoke conjunctivitis as a side effect [129].

Lebrikizumab is a new monoclonal antibody that targets IL-13 which inhibits the formation of the signaling complex of the IL-13Rα1/IL-4Rα receptor in adult patients with moderate to severe atopic dermatitis [100,101,130,131].

When combined with topical corticosteroids and given subcutaneously every 4 weeks to patients with AD, the monoclonal IL-13 antibody lebrikizumab outperformed the control group. At week 20, dose-dependent responses for the percentage change from the baseline pruritus VAS were noted [100]. In a phase 2b, placebo-controlled, double-blind trial, as early as day 2 in the higher-dosage group, the humanized high-affinity IL-13 antibody lebrikizumab significantly reduced the severity of AD and the intensity of pruritus [101]. In a preclinical study regarding the role of IL-13 and its blockage in pruritus, it was shown that IL-13 is a powerful amplifier of neural responses to various itch stimuli, and it is therefore likely that the neuroimmune axis plays a role in the development of inflammatory skin diseases linked to chronic itch [132].

### 3.5. IL-31

IL-31 belongs to the IL-6 cytokine family [118]. IL-4, a significant atopic dermatitis mediator, stimulates IL-31 production in human lymphocytes via activating the JAK-STAT signaling pathway. IL-33, which is secreted by keratinocytes in response to injury, works in concert with IL-4 to increase the production of IL-31 via the NF-kB pathway [133,134]. A heterodimeric receptor complex made up of the oncostatin M receptor beta (OSMRβ) and the IL-31 receptor alpha (Il-31RA) is responsible for transmitting the IL-31 signal [135]. When the IL-31 receptor is activated, either directly or indirectly, at least four signaling pathways—AKT/PI3K, JAK-STAT, MAPK (ERK1/2, p38, JNK), and NF-kB—are activated in a cell-dependent manner [135,136,137,138,139]. IL-31 activity in keratinocytes is mostly regulated by STAT3 phosphorylation, while various IL-31RA isoforms may have an impact on this activity [136]. IL-31RA has a changeable domain at the end terminal that can be either long or short. The strongest STAT3 activity is induced by the long isoforms 1 and 2 [133]. Similar to neuron growth factor, IL-31 stimulates neural development in mice by phosphorylating STAT3 [42]. The OSMR subunit of the Il-31 receptor may activate the remaining signaling pathways even if the IL-31RA subunit is what causes STAT3 activation [136,140]. Independent of p38, IL-31-induced pruritus is brought on by ERK phosphorylation in the primary afferent neurons of the dorsal root ganglia. Short and long versions of IL-31RA undergo identical ERK activation when activated by IL-31 [121]. Although pruritus provoked by IL-31 may occur independently of mast cells, IL-4 causes overexpression of IL-31 in lymphocytes and mast cells [121]. When IL-31 is injected or overexpressed pruritus is provoked and the neural growth of neurons with a small diameter is selectively promoted which activates several of the same genes as a nerve growth factor [140]. In mice, calcium influx-induced pruritus is mediated through transient receptor potential cation channel 1 and transient receptor potential vanilloid 1 (TRPV1). In TRPA1-deficient mice, pruritus induced by IL-31 is significantly attenuated [121]. Studies have shown that IL-31 is overexpression in transgenic mice causes primary alopecia, severe pruritus, and peripheral lymphadenopathy independent of mast cells [135]. Upregulation of IL-31 has been shown to cause an atopic dermatitis-like phenotype in mouse models [137].

Nemolizumab is a newly discovered humanized monoclonal antibody against receptor A of IL-31 [136]. Nemolizumab may be useful in reducing atopic dermatitis-related pruritus, according to several trials. Nemolizumab displayed a significant improvement in the EASI score and pruritus at a dose of 30 mg every four weeks in a phase 2b trial when combined with TCS [97]. The efficacy of nemolizumab in reducing pruritus related to AD was validated further in a Japanese randomized trial that investigated 215 patients aged 13 years or older with AD and moderate to severe pruritus [98]. In this phase 3 trial, the Japanese AD group showed a higher reduction in pruritus score at a dose of 60 mg of nemolizumab combined with TCS every four weeks [98]. In two long-term phase III studies, in the nemolizumab group, the pruritus Visual Analogue Scale scores decreased by 66% and the EASI score by 78% [99]. The strong antipruritic effect of nemolizumab is promising [141,142].

### 3.6. IL-6

Interleukin (IL)-6 is a proinflammatory cytokine, that exerts a great variety of activities during inflammatory illness, host infection, oncogenesis, and hematopoiesis. It was first identified as a B cell differentiation factor. Additionally, it has anti-inflammatory properties [143,144]. The IL-6 receptor (IL-6R) is composed of two functional chains: the IL-6R alpha (IL-6Ra) and the 130 kD glycoprotein 130 (gp130), a non-ligand binding chain capable of signal transduction, which mediates gene activation and a variety of biological actions [145]. IL-6 binds to both membrane-bound and soluble receptors, by activating three distinct but interconnected signaling pathways [146,147]. Immune response, specifically T cell proliferation and differentiation, as well as B cell terminal differentiation, are regulated by IL-6. IL-6 is a key modulator of the hepatic acute phase response and also activates macrophages and osteoclasts. IL-6 promotes the production of metalloproteinase and vascular endothelial growth factor (VEGF) in conjunction with tumor necrosis factor (TNF) alpha and IL-1. Additionally, transforming growth factor (TGF) beta and IL-6 play a part in the maturation of regulatory T cells. Similar to IL-1, IL-6 is crucial for systemic and local inflammation, which is frequently accompanied by symptoms like fever, exhaustion, and anorexia as well as alterations in acute phase proteins in the plasma (e.g., C-reactive protein and fibrinogen) [146,147,148]. Mast cells (MCs) and activated T cells produce IL-6, and it has been shown that is highly expressed in the skin and T cells of AD patients [149,150,151].

Serum naturally contains IL-6 in levels of a picogram per milliliter (pg/mL), a glycoprotein from the IL-6 family that is implicated in both health and sickness. The release of IL-6 by the skeletal muscle during exercise or even following multiple injuries can cause IL-6 levels to rise in practically any inflammatory condition [152].

Tocilizumab, a human recombinant monoclonal antibody against the IL-6 receptor that is administered intravenously, was effective in three patients with severe AD but it was also related to bacterial superinfection [153]. There are reports that IL-6 is associated with prurigo nodularis [154] and calcium phosphate-induced pruritus [155].

### 3.7. Endothelin-1 (ET-1)

Endothelin-1 (ET-1), a 21-amino-acid peptide, is a potent direct vasoconstrictor, also known as a histamine-independent pruritogen that may stimulate the proliferation of fibroblasts and vascular smooth cells [156]. Several different cell types, including endothelial cells, and immunological cells such as dendritic cells, keratinocytes monocytes, neurons, and macrophages, express this 21-amino-acid peptide [157,158]. ET-1 mainly acts via two receptors, ETAR and ETBR. Only the ETA receptor has a role in pruritus transmission. A study showed that the neural peptidase ECE-1 controls pruritus caused by ET-1 in mice and humans, acting as a negative regulator of itch on sensory nerves [159]. Additionally, another study found increased ET-1 expression in the cutaneous lesions of AD patients [160]. Elevated serum ET-1 levels are associated with itch intensity, serum IgE levels, and the severity of AD [161]. Another research group demonstrated that the expression of ET-1 is elevated in the skin and serum of prurigo nodularis patients [162]. In a mouse-induced AD model, pruritus and dermatitis were both improved with the use of a nonselective ETAR and ETBR antagonist, bosentan [158].

### 3.8. Neurotrophins (NTs)

Neurotrophins (NTs) are proteins that induce the development, function, and survival of neurons. Additionally, they boost neurons’ plasticity and damage tolerance and inhibit pathways that lead to apoptosis, enabling nerve cells to survive under conditions of elevated oxidative stress. NTs include brain-derived neurotrophic factor (BDNF), nerve growth factor (NGF), NT3, and NT4/5 [163]. It has been demonstrated that serum levels of NGF are elevated in AD patients [164,165]. Given the fact that other studies have not shown any correlation between AD severity and NGF levels, this might suggest that NGF might have a local effect on AD pathophysiology [166,167]. A recent study reported raised levels of NT-4 in a group of patients with chronic kidney-induced pruritus. Additionally, there was no connection between serum NT-4 concentrations and the intensity of pruritus [168]. A topical tropomyosin-receptor kinase A (TrkA) inhibitor, pegcantratinib CT327, has been shown to significantly alleviate pruritus in psoriasis patients [169].

### 3.9. Neuropeptides

Neurons produce and release neuropeptides, which are chemical messengers consisting of short chains of amino acids. Normally, neuropeptides attach to G protein-coupled receptors (GPCRs) to influence the nervous system’s function and other tissues like the heart, muscles, and gut. The biggest and most diversified class of signaling molecules in the nervous system, neuropeptides, are known to number over 100. Common neuropeptides that are released by cutaneous nerve terminals are substance P (SP), vasoactive intestinal peptide (VIP), and calcitonin gene-related peptide (CGRP). In particular, SP and CGRP have a well-established role in the pathogenesis of AD [170].

The peptide neurotransmitter called substance P (SP) is released by sensory neurons. Its receptor is found in a wide variety of cell types and tissues, and it has the ability to control inflammation, cytokine release, and immune cell proliferation. It has direct antibacterial activity and shares common physical and chemical characteristics with AMPs [171]. SP signals mainly via the neurokinin 1 receptor (NK1R) [172]. Additionally, SP binds to MRGPCRs (Mas-related G protein-coupled receptors) that play an important role in pruritus signaling and nociception [173]. One study showed that AD patients have elevated serum SP levels and elevated expression of SP and NKIR+ in lesional skin [174], but a different study did not find any correlation between AD and serum SP levels [175]. One study found the use of a neurokinin receptor 1 (NKR1) antagonist, aprepitant, effective in treating pruritus [176], but another study did not show any effect as an adjuvant treatment to topical corticosteroids (TCS) [177]. In a phase 3 clinical trial (EPIONE), tradipitant (VLY-686) a new NK-1R antagonist was beneficial for treating pruritus and aiding sleep in patients with mild AD [103].

Calcitonin gene-related peptide (CGRP) is mainly found in primary afferent sensory fibers. Human CGRP comes in two different forms, CGRP- and CGRP-, with CGRP- being the main type that is present in the central and peripheral nervous systems. LCs, monocytes, and keratinocytes are a few examples of cell types that can release CGRP in addition to cutaneous nerve endings [178]. Inflammatory cytokines can be produced by keratinocytes and MC degranulation when CGRP is present [178,179]. Most significantly, CGRP steers LCs toward a Th2 pole, which promotes atopic skin disease [180]. Patients with AD have been observed to have higher levels of serum CGRP and lesioned CGRP+ nerve fibers [181,182]. CGRP hyposecretion and SP hypersecretion in the skin has been observed in an AD mouse model, [183].

In patients with AD, elevated serum levels of neuropeptide Y (NPY) [182] and increased serum VIP levels are associated with pruritus [184,185]. VIP inhibits Th1 differentiation while promoting Th2 cell survival and differentiation [186]. NPY stimulates Th2 differentiation, boosts IL-4 production, and is necessary for type 2 responses [187]. Patients with AD have been found to have increased expression of GRP in the skin and elevated serum levels of GRP, and there is a link between the intensity of their pruritus [188,189]. A study demonstrated that levels of GRP expression in the skin of AD patients were related to the severity of AD and pruritus intensity [188].

### 3.10. Toll-like Receptors (TLRs)

Toll-like receptors (TLRs) are transmembrane pattern recognition receptors (PRRs) that are present on and within innate immune system cells, including dendritic cells, monocytes, macrophages, epithelial cells, and neutrophils. TLRs can recognize a range of pathogen-associated molecular patterns (PAMPs) and damage-associated molecular patterns (DAMPS), such as proteins and nucleic acids found in microbial cell walls [190]. High susceptibility to AD is connected with TLR2 and TLR4 polymorphisms [191]. Additionally, TLR2 activation may lead to increased chemokine mRNA production, which could aid in AD development [192]. In AD patients, high expression of TLR3 in the epidermis is correlated with AD severity and hydration [193]. Additionally, TLR3 can activate a variety of pruritogens in keratinocytes such as ET-1, TSLP, and IL-33 [194,195]. In a dry-skin mouse model, the knockdown of TLR3 in dorsal root ganglia (DRG), showed exacerbation of pruritus [196]. Although TLR7 is related to non-histamine pruritus [197], imiquimod, which is a TLR7 agonist, showed a TLR7-independent and TRV1-dependent pruritus pathway [198].

A change in the diversity of the microbiota affects how AD develops and progresses, for example, a decrease in microbiome diversity is correlated with disease severity, especially in the lesional skin of AD. Current research suggests the possibility that the microbiome may influence AD itching through interactions between the gut, skin, and brain [199].

Acute itch behavior may be lessened by antibiotic-induced gut microbiota depletion, which may be related to lessened trigeminal ganglia (TG) neuronal activity [200].

### 3.11. Janus Kinase (JAK)/Signal Transducer and Activator of Transcription (STAT)

The Janus kinases (JAKs) are protein tyrosine kinases (TYKs) that bind to transmembrane type 1 and type 2 cytokine receptors and mediate cellular responses to numerous cytokines and growth factors; these mediators are important in immune defense and in immune-mediated disease [201]. Persistent pruritus in AD leads to scratching, which in turn triggers STAT3 activation in astrocytes of the spinal dorsal horn. Activation of lipocalin (LCN) 2 by astrocytes sensitizes a pruritus-processing neural network made up of GRPR1 spinal dorsal horn neurons, which in turn causes chronic pruritus and the cycle of itch and scratch [35,202] (Figure 2).

Janus kinases (JAKs) are a family of kinases that include JAK1, JAK2, JAK3, and tyrosine kinase 2 (TYK2); JAKs mediate cytokine signaling by activating signal transducer and activator of transcription (STAT) transcription levels. The JAK/STAT pathway is activated in a variety of disease states [203].

Cytokines including IL-31, IL-4, IL-13, and TSLP are crucial for pruritus in AD transmit their signals into the cells via JAK-1 and JAK-2 [202]. In animal studies, JAK-1/2 inhibition or deletion greatly decreased the itch signals brought on by these mediators [119]. When the oral JAK 1/3 inhibitor tofacitinib greatly decreased pruritus in elderly patients with chronic pruritus of unknown cause, it was demonstrated that JAKs play a crucial role in human pruritus. In AD, inflammation and pruritus are influenced by cytokines such as IL-31, IL-4, IL-13, TSLP, and IL-5. A new therapy option is made possible through the possibility of inhibiting JAK-1 and JAK-2 with specific JAK inhibitors [119].

Abrocitinib, an oral selective JA-1 inhibitor, has been proven to significantly reduce pruritus in AD [81,82,84]. In a randomized, phase 2b, double-blinded, placebo-controlled trial in adult patients with AD, abrocitinib (200 mg, 100 mg, 30 mg, or 10 mg) or a placebo were given to participants at random once a day for 12 weeks. In comparison to the placebo group, differences in pruritus were seen in the 200 mg and 100 mg groups [84]. In a further double-blind, randomized phase 3 trial (JADE MONO-1), with patients with moderate to severe AD and more than 12 years old, abrocitinib 200 mg, abrocitinib 100 mg, or a placebo was given to patients at random (2:2:1) once a day for 12 weeks. Between the start of the treatment and week 12, a considerable, quick (i.e., within two days) reduction in the severity of pruritus and other atopic dermatitis symptoms was noticed [81]. In a phase 3 trial, in terms of itch response at week two, abrocitinib 200 mg was superior to dupilumab, but the 100 mg dose was not. For the majority of other important secondary end-point comparisons, neither abrocitinib dose substantially varied from dupilumab at week 16 [83].

Baricitinib, an oral JAK inhibitor that reversibly inhibits JAK1 and JAK2 and moderately inhibits TYK2 has been shown to reduce pruritus in AD patients [85,86].

Furthermore, upadacibitinib, an oral JAK-1 inhibitor, has effectively reduced pruritus in AD [87,88]. With each upadacitinib dosage regimen of 7.5, 15, or 30 mg once daily, a significant improvement in pruritus from the baseline to week 16 was also observed compared to the placebo in a clinical trial [87].

Topical delgocitinib is an investigational JAK inhibitor that blocks tyrosine kinase 2, JAK1, JAK2, and JAK 3. When given to adult Japanese patients with moderate to severe atopic dermatitis for up to 28 weeks, delgocitinib 0.5% ointment, a new topical Janus kinase inhibitor, improved clinical signs and symptoms with a positive safety profile [89]. Nakagawa H. et al. 2020 JAAD. At week one, the delgocitinib group had a lower pruritus Numerical Rating Scale (NRS) than the vehicle group, and this difference persisted over time in adults and children [89,204]. Tofacitinib reversibly inhibits JAK1 and JAK3 in vitro and to a lesser extent inhibits JAK2 and TYK2 [89]. Tofacitinib, by inhibition of JAK3, inhibits cytokines IL-2, IL-4, IL-7, IL-9, IL-15, and IL-21; and through JAK1 inhibition, inhibits IL-6, type 1 IFNs, and IFN-gamma. In a phase 2a, randomized trial, the effectiveness of topical tofacitinib for the treatment of atopic dermatitis was assessed. In this trial, tofacitinib 2% ointment or placebo was applied twice daily for a period of four weeks on 69 adult patients with clinically stable, mild to moderate atopic dermatitis. Significant improvements in pruritus in the tofacitinib group were observed by day two [91,205].

A topical version of the drug ruxolitinib, which is a strong and specific inhibitor of Janus kinase JAK1 and JAK2, is available as ruxolitinib cream 1.5% (OPZELURATM) that has been used topically in atopic dermatitis [206]. Ruxolitinib cream 1.5%, when used twice daily for eight weeks, significantly reduced indices of disease severity, pruritus, and sleep disturbance in two identically constructed, international, phase 3 studies in patients with mild to severe AD aged 12 years [92,206,207].

## 4. Conclusions

The most distressing symptom of AD, regardless of its severity level, is chronic pruritus. It affects patients’ quality of life significantly and adversely affects their wellness. Our findings suggest that one of the major players in the pathophysiology of AD is pruritogen. The neurocutaneous interaction in chronic pruritic skin disorders is currently being partly elucidated thanks to significant research on the pathophysiology of pruritus. This aids the development of AD-specific treatments. The development of biologics that target IL-4, IL-13, IL-31, and other molecules has produced promising results. The development of new and efficient medicines for atopic dermatitis (AD) may provide a great variety of options for treatment for AD patients. Additionally, new AD therapies will allow us to adapt our treatment plan to our AD patients’ current and future needs.

## Figures and Tables

**Figure 1 jcm-12-02091-f001:**
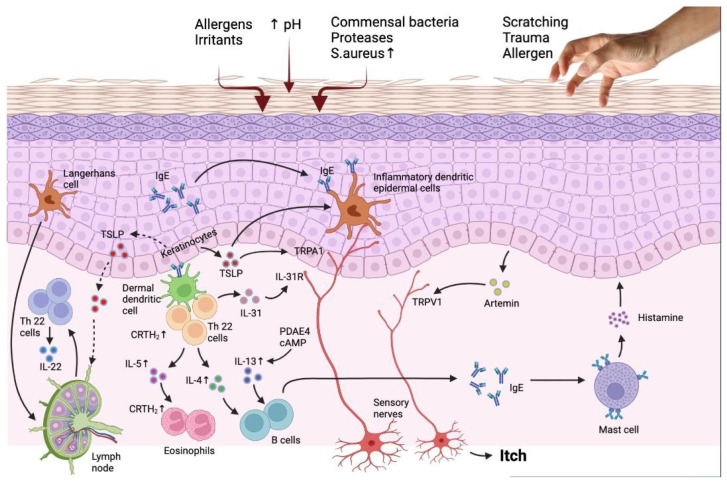
The interaction of the immune system and epidermis in AD patients reveals the targets for novel treatment. Barrier flaws encourage the entry of epicutaneous antigens that interact with dermal dendritic cells and epidermal Langerhans cells, activating the immune system, especially the TH2 and TH22 signaling pathways. TH2 cell migration into the skin is facilitated by the prostaglandin D2 receptor CRTH2, which is found on a number of inflammatory cells. Peripheral eosinophils and mast cells are stimulated by TH2 cytokines, such as IL-4, IL-3, and IL-5, which also cause IgE class change. Increased PDE4 also boosts TH2 cytokine expression. IL-33, which stimulates epidermal hyperplasia and is abundant in individuals with chronic diseases, is produced by TH22 cells. The reduced production of barrier proteins and barrier impairment caused by TH2 and TH22 cytokines are also thought to enhance the risk of infection. Flares result in an increase in S aureus and a decrease in bacterial diversity. Increased IL-31, a TH2 cytokine, is linked to itching but not inflammation. It is assumed that TSLP synthesis by keratinocytes causes itch by directly activating transient receptor potential A1 (TRPA1) receptors in cutaneous sensory neurons in addition to inducing a TH2 response. AhR and H4R, are more recent targets that control IL-31 expression by being expressed on keratinocytes and TH2 cells, respectively. Adapted from: Paller et al. [35].

**Figure 2 jcm-12-02091-f002:**
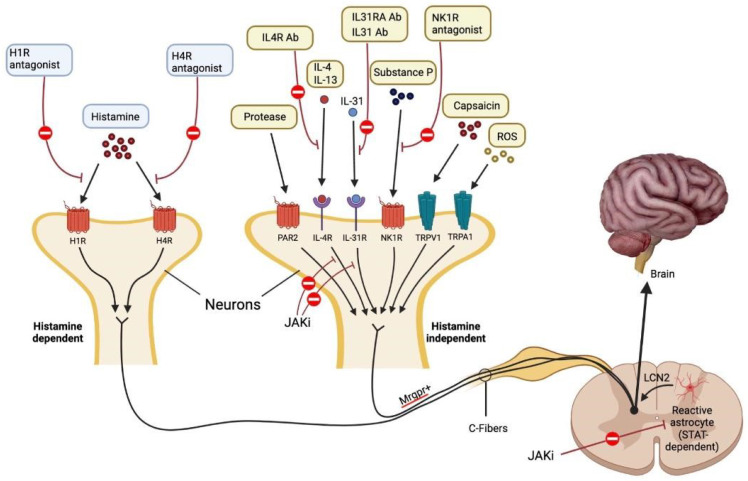
Mechanisms that cause itch. Each pruritogen involved in AD has a unique receptor that causes itching. It is believed that people with AD experience pruritus through both H1R1 histamine-dependent and Mas-related G protein-coupled receptor (Mrgpr)1 histamine-independent signaling pathways. Lipocalin 2 (LCN2), which is produced by STAT3-dependent reactive astrocytes in the spinal dorsal horn and sensitizes a network of GRPR1 neurons involved in the processing of itch, causes chronic pruritus. Each antipruritic medication blocks a different pruritic route, as depicted in the diagram. JAK-STAT signaling causes IL-4, IL-13, and IL-31 to perform their specific tasks. Protease-activated receptor 2 is abbreviated as PAR2. Adapted from: Paller et al. [35].

**Table 1 jcm-12-02091-t001:** An overview of important pruritogenic mediators in atopic dermatitis (AD).

Mediator	Origin	Overall Mechanism	Receptor
Histamine	Mast cells (MCs), basophils	Histamine provokes pruritus through H1R and H4R.	It has many activities via four receptors (H1R–H4R).
Platelet Activating Factor (PAF)	MCs, eosinophils, basophils, neutrophils and epithelial cells	PAF boosts the immune response by causing immune cells like eosinophils and MCs to degranulate, and trigger chemotaxis and adhesion. PAF injection intradermally might cause wheal and irritation.	Platelet-activating factor receptor (PAF-R)
Protease and (PARs) Protease-Activated Receptors	Keratinocytes and immune cells	Through neurons that express BLT1, the activation of PAR-2 on keratinocytes causes the release of LTB4, which causes itching.	PAR1, PAR2, PAR3, and PAR4. In AD, the functional roles of PAR2 have been characterized in greater detail.
Thymic Stromal Lymphopoietin (TSLP)	TSLP is a cytokine produced by keratinocytes.	Raised levels of TSLP are expressed in keratinocytes of lesional skin and serum of patients with atopic dermatitis.	IL-7Rα, TSLPR
IL-33	It functions as an alarmin and is swiftly released from the keratinocyte nucleus in response to pathogen provocation or epidermal barrier disruption.	IL-33 communicates via IL-33 Receptor (ST2)	ST2
IL-4	CD4+ T cells, basophils, eosinophils	Enhances IL-31/IL-31 receptor α signaling. It amplifies Th2 inflammation and increases the production of IgE.	IL-13 receptor α1 chain, IL-13 receptor α2 chain
IL-13	CD4+ T cells, basophils, eosinophils	The ability IL-13 IS to sensitize sensory nerves to itch by reducing the sensitivity thresholds to other pruritogenic stimuli	IL-4 receptor type II (IL-4RII)
IL-31	Th2 cells	Signals via IL-31RA, Interleukin-31 receptor α chain and OSM Receptor (OSMR) β chain.	Oncostatin M receptor beta (OSMRβ) and the IL-31 receptor alpha (Il-31RA
IL-6	It is released by activated T cells and mast cells (MCs).	Mast cells (MCs) and activated T cells produce IL-6 and it has been shown that is highly expressed in the skin and T cells of AD patients.	IL-6R alpha (IL-6Ra) and the 130 kD glycoprotein 130 (gp130) non-ligand binding chain
Endothelin-1 (ET-1)	It is produced by vascular endothelial cells.	Serum ET-1 levels are elevated and associated with itch intensity, serum IgE levels, and the severity of AD.	ETAR, ETBR
Neurotrophins (NTs)	Keratinocytes in the skin	Hyperinnervation, peripheral sensitization, and pruritus in atopic dermatitis.	TrkA, TrkB, and TrkC (tyrosine kinases)

**Table 2 jcm-12-02091-t002:** Summary of new atopic dermatitis treatments’ effectiveness in reducing pruritus.

Drug	Therapeutic Target	Mode of Administration	Type of Study	Dosing and Intervention	Duration in Weeks	Number of Participants (Medication/Placebo)	Age of Participants	Outcome Measure	Effect on Pruritus
Abrocitinib	JAK1 selective inhibitor	oral							
Simpson E.L. et al. *Lancet* 2020 [81]	JAK1 selective inhibitor		A multicenter, double-blind, randomized phase 3 trial (JADE MONO-1)	Abrocitinib 200 mg and 100 mg per os daily monotherapy and placebo	12 week study	Total 387 participants, abrocitinib 100 mg daily (*n* = 156), abrocitinib 200 mg (*n* = 154), placebo (*n* = 77)	Adolescents and adults	Peak Pruritus Numerical Rating Scale [PP-NRS] score, ranges from 0 to 10	PP-NRS response at week 8 Abrocitinib 100 mg: 50/147 (34%). Percentage difference compared to placebo (95% CI) 20·0 (7·4 to 32·7), abrocitinib 200 mg: 88/147 (60%), percentage difference compared to placebo (95% CI) 45·3 (32·7 to 57·8), a significant, rapid (i.e., within 2 days) reduction in pruritus severity and other atopic dermatitis symptoms were also observed between treatment initiation and week 12. The median time to PPNRS response was 84⋯0 days (IQR 10⋯0—not evaluable [NE]) in the abrocitinib 100 mg group, 14 · 0 days (6⋯0–84⋯0) in the abrocitinib 200 mg group, and 92⋯0 days (29⋯0—NE) in the placebo group.
Silverberg J.I., *JAMA Dermatol.* 2020 [82]			Phase 3, double-blinded, placebo-controlled, parallel-group randomized clinical trial	Abrocitinib 200 mg and 100 mg per os daily monotherapy and placebo	12-week study	391 in total, placebo (*n* = 78), abrocitinib 100 mg (*n* = 158), abrocitinib 200 mg (*n* = 155)	12 years old and older, adolescents and adults	PP-NRS, Peak Pruritus Numerical Rating Scale, ranges from 0 to 10; PSAAD, Pruritus and Symptoms Assessment for Atopic Dermatitis	Response was achieved in the 200 mg group by 35.3% of patients at week 2, 52.8% at week 4, and 55.3% at week 12 and in the 100-mg group by 23.1% at week 2, 33.4% at week 4, and 45.2%at week 12 compared to 11.5% of the placebo group at week 12.
Bieber T., *N. Engl. J. Med*., 2021 [83]			Phase 3, double-blind trial	(2:2:2:1 ratio) Abrocitinib 200 mg or 100 mg orally once daily, dupilumab 300 mg subcutaneously every other week (after a loading dose of 600 mg), or placebo plus topical therapy.	12-week study	838 in total, abrocitinib 200 mg (*n* = 226), abrocitinib 100 mg (*n* = 238), dupilumab (*n* = 243), placebo (*n* = 131)	18 years of age or older, adult patients	PP-NRS	Abrocitinib 200 mg dose, but not the abrocitinib 100 mg dose, was superior to dupilumab with respect to itch response at week 2.
Gooderham M.J. et al. *JAMA Dermatol.* 2019 [84]			Phase 2b, randomized, double-blinded, placebo-controlled, parallel-group trial	(1:1:1:1:1) participants received abrocitinib (200 mg, 100 mg, 30 mg, or 10 mg) or placebo once daily	12-week study	In total = 267 participated, but 263 were included in the full analysis. Placebo (*n* = 55), abrocitinib 10 mg (*n* = 49), abrocitinib 30 mg (*n* = 50), abrocitinib 100 mg (*n* = 55), abrocitinib (*n* = 54)	18 to 75 years, adults	Pruritus NRS score (0–10)	At week 12, significant reductions in pruritus NRS scores were observed in the 200 mg (LSM difference from the placebo, –25.4%; *p* = 0.003) and 100 mg (−20.7%; *p* = 0.02) groups compared with placebo. Odds ratio abrocitinib 200 mg: 5.11 (2.43 to 10.77), abrocitinib 100 mg: 2.84 (1.40 to 5.76).
Baricitinib	Janus kinase 1/Janus kinase 2 inhibitor	Oral							
Simpson E.L. et al. *JAAD*, 2021 [85]	Janus kinase 1/Janus kinase 2 inhibitor		Phase 3 trial (BREEZE-AD5/NCT03435081)	1:1:1 once-daily placebo or baricitinib 1 mg or 2 mg.	16-week study	440	Adults	Itch Numerical Rating Scale (NRS)	The proportion achieving 4-point improvement on the itch, NRS was 6% for placebo, 16% for baricitinib 1 mg, and 25% for baricitinib. 2 mg (*p* < 0.001)
Guttman-Yassky E. et al. *JAAD* 2019 [86]			Phase 2, randomized, double-blind, placebo-controlled study	Placebo plus topical corticosteroids (TCS), baricitinib 2 mg plus TCS, baricitinib 4 mg plus TCS	16-week study	124 in total, placebo plus topical corticosteroids (TCS) *n* = 49, baricitinib 2 mg plus TCS (*n* = 37), baricitinib 4 mg plus TCS (*n* = 38)	Adults	NRS, SCORAD patient-reported items of pruritus	Baricitinib plus TCS showed early and significant reduction in cutaneous inflammation and pruritus.
Upadacitinib	Janus kinase (JAK)1-selective inhibitor	Oral							
Guttman-Yassky E. et al. 2020 *JACI* [87]			Phase 2b, double-blind, placebo-controlled, parallel-group, dose-ranging portion	1:1:1:1, placebo or extended-release upadacitinib (manufactured by the study sponsor) 7.5, 15, or 30 mg once daily (QD) by mouth	16-week study	167 patients were randomized, placebo treated (*n* = 40), upadacitinib 7.5 mg (*n* = 42), upadacitinib 15 mg (*n* = 42), upadacitinib 30 mg (*n* = 42),	Adults, 18 to 75 years	NRS (0–10)	Each upadacitinib dose level was significantly superior to the placebo. Patient assessment of pruritus (improvement in NRS and achievement of NRS reduction > 4) at week 16.
Reich K. et al. *Lancet* 2021 [88]			Randomized, double-blind, placebo-controlled, phase 3 trial (AD Up)	(1:1:1) to receive upadacitinib 15 mg, upadacitinib 30 mg, or placebo once daily, all in combination with topical corticosteroids (TCS)	16-week study	785 adults plus 116 adolescents were randomized, upadacitinib 15 mg + TCS (*n* = 300), upadacitinib 30 mg + TCS (*n* = 297), placebo + TCS (*n* = 280 completed of initial 304)	Adolescents and adults, age above 12 years old	Weekly average worst pruritus Numerical Rating Scale score, NRS (0–10)	Significant reduction in pruritus in both upadacitinib groups.
Delgocitinib	All JAKS	Topical use, delgocitinib 0.25% and 0.5% ointment							
Nakagawa H. et al. 2020 *JAAD* [89]		Delgocitinib 0.5% ointment	2:1 ratio to delgocitinib 0.5% ointment or vehicle ointment	2:1 ratio to delgocitinib 0.5% ointment or vehicle ointment	1st part: 4 weeks, 2nd part: 24 weeks	158 patients delgocitinib (*n* = 106), vehicle (*n* = 52)	16 years or older	Pruritus NRS scores across parts 1 and 2.	Pruritus NRS scores were significantly improved in the delgocitinib group compared with those in the vehicle group. The pruritus NRS score in the delgocitinib group was lower than in the vehicle group at week 1, which was maintained over time.
Nakagawa H. et al., 2021, *JAAD* [90]		In part 1, delgocitinib 0.25% ointment. In part 2, delgocitinib ointment at a concentration of 0.5%.	1:1 double-blind study	Part 1 of this study was a 4-week double-blind period in which Japanese patients ages 2 through 15 years were randomized in a 1:1 ratio to delgocitinib 0.25% ointment or vehicle ointment.		137 in total, delgocitinib (*n* = 69), vehicle (*n* = 68)	Children, patients ages 2 through 15 years	NRS	Improvement in pruritus scores for the delgocitinib group
Tofacitinib	JAK 1/3	Topical							
Bissonnette R. et al., 2016, *BJD* [91]		Topical	4-week, phase IIa, randomized, double-blind, vehicle-controlled study (NCT02001181)	1:1 to 2% tofacitinib or vehicle ointment twice daily	4-week study	Tofacitinib 2% bd (*n* = 35), vehicle (*n* = 34)	Adults	Percentage change from baseline (CFB) in patient-reported pruritus	Significant improvements in pruritus were observed by day 2.
Ruxolitinib **(RUX)**	JAK 1/2	Topical 0.75% RUX cream and 1.5% RUX cream							
Papp K, et al., 2021, *JAAD* [92]			Two phase 3 studies	1:1:1 vehicle, 0.75% RUX cream and 1.5% RUX cream	8-week study	TRuE-AD1: 631 patients were randomized (vehicle, *n* = 126; 0.75% RUX, *n* = 252; 1.5% RUX, *n* = 253). In total, 558 (88.4%) completed the 8-week study TRuE-AD2 comprised 618 randomized patients (vehicle, *n* = 124; 0.75% RUX, *n* = 248; 1.5% RUX, *n* = 246). In total, 561 (90.8%) patients, respectively, completed the 8-week study.	Patients aged 12 years or older	NRS (0–10)	Significant itch reductions versus vehicle were reported within 12 hours of first application of 1.5% RUX (*p* < 0.05).
Tralokinumab	IL-13	Subcutaneous (sc) injection							
Wollenberg A. et al., 2019, *JACI* [93]			Phase 2b study (NCT02347176) https://clinicaltrials.gov/ct2/show/NCT02347176 (accessed on 2 February 2023)	1:1:1:1 to receive subcutaneous tralokinumab 45, 150, or 300 mg, or placebo, every 2 weeks for 12 weeks with concomitant topical glucocorticoids.	12-week study	204 adults, placebo (*n* = 51), tralokinumab 45 mg (*n* = 50), tralokinumab 150 mg (*n* = 51), tralokinumab 300 mg (*n* = 52)	Adults	Pruritus Numerical Rating Scale (7-day mean) scores P-NRS (0–10)	Participants demonstrated improvements from baseline to week 12 in pruritus Numerical Rating Scale (7-day mean) scores versus those receiving placebo when receiving 45 or 300 mg of tralokinumab. These improvements were observed from week 1 onward for all tralokinumab doses and maintained beyond week 12.
Wollenberg A. et al., 2020, *BJD* [94]			Phase III trials, ECZTRA 1 and ECZTRA 2	3:1 to subcutaneous tralokinumab 300 mg, after a 600-mg loading dose on day 0, or a placebo every other week for 16 weeks. After a 16-week initial treatment period, tralokinumab-treated patients who achieved the prespecified criteria for clinical response were rerandomized 2:2:1 to tralokinumab 300 mg every 2 weeks (Q2W) or every 4 weeks (Q4W), or placebo for a 36-week maintenance treatment period.	16 weeks initially, 52-week study	ECZTRA 1: placebo (*n* = 199), tralokinumab patients (*n* = 603), ECZTRA 2: placebo (*n* = 201), tralokinumab patients (*n* = 593)	Adults	P-NRS (0–10)	Early improvements in pruritus were observed.
Silverberg, J.I. et al. *Br. J. Derm.* 2021 [95]			Phase III trial	2:1 to subcutaneous tralokinumab 300 mg or placebo every 2 weeks (Q2W) with TCS. At week 16 tralokinumab patients were rerandomized 1:1 to tralokinumab Q2W or every 4 weeks (Q4W), with TCS as needed, for another 16 weeks.	16-week study	All participants = 380, placebo Q2W + TCS (*n* = 127), tralokinumab Q2W + TCS (*n* = 253)	≥18 years of age adults	Worst daily pruritus Numerical Rating Scale (NRS)	Greater reduction in weekly average of worst daily pruritus was observed in the tralokinumab arm.
Gutermuth J. et al. *Br. J. Dermatol.* 2022 [96]			Parallel, randomized, double-blind, placebo-controlled, phase III trial	1:1 to subcutaneous tralokinumab 300 mg or placebo every 2 weeks plus TCS as needed	26-week study	277 patients	Adults	Worst daily pruritus Numerical Rating Scale (NRS)	Improvement in pruritus in patients treated with tralokinumab.
Nemolizumab	IL-31 receptor α subunit	Subcutaneous (sc) injection							
Silverberg J.I. et al. *JACI*, 2020 [97]			Phase 2B randomized study	Nemolizumab (10, 30, and 90 mg) subcutaneous injections every 4 weeks versus placebo, with topical corticosteroids (TCS)	24-week study	Total = 226, Placebo *n* = 57, nemolizumab 10 mg *n* = 55, nemolizumab 30 mg *n* = 57, nemolizumab 90 mg *n* = 57	Adults	Weekly average pruritus NRS score (0–10)	All doses of nemolizumab were associated with a rapid decrease in pruritus scores, with statistically significant differences from placebo starting as early as week 1. By week 2, scores with all nemolizumab doses were greater than those with placebo (*p* < 0.001).
Kabashima K. et al. *NEJM*, 2020 [98]			Double-blind, phase 3 trial	2:1 ratio to receive subcutaneous nemolizumab (60 mg) or placebo every 4 weeks until week 16, with concomitant topical agents.	16-week study	In total 215 patients, nemolizumab *n* = 143, placebo *n* = 72	Aged 13 years or older, a body weight of 30.0 to 120.0 kg	Pruritus VAS score (0–100), NRS (0–10), itch score (0–4)	At week 16, the least-squares mean percent change from baseline in the pruritus VAS score (primary end point) was −42.8% in the nemolizumab group and −21.4% in the placebo group.
Kabashima K. et al. *BJD*, 2022 [99]			Two long-term phase III studies	Nemolizumab 60 mg every 4 weeks (Q4W) was administered subcutaneously, concomitantly with topical treatments. Study-JP01 patients received double-blind nemolizumab or placebo for 16 weeks, and then entered a 52-week extension period in which all patients received nemolizumab (nemolizumab/nemolizumab and placebo/nemolizumab groups). Study-JP02 patients received nemolizumab for 52 weeks. Both studies included an 8-week follow-up period.	16 and 54 weeks	Study-JP01 nemolizumab/nemolizumab and placebo/nemolizumab, and Study-JP02 nemolizumab groups comprised 143, 72 and 88 patients, respectively.	aged ≥ 13 years, with a bodyweight of ≥30_0 kg	Pruritus VAS score (range 0–100), five-level itch scale (range 0–4), the pruritus Numerical Rating Scale (NRS, range 0–10),	Improvement in pruritus in patients treated with nemolizumab.
Tezepelumab	anti-thymic stromal lymphopoietin monoclonal antibody, TSLP	Subcutaneous injection							
Simpson EL. et al., 2019, *JAAD* [80]			Phase 2a study (NCT02525094)	1:1 to subcutaneous tezepelumab 280 mg or placebo plus TCS every 2 weeks	16-week study	111 patients in total, placebo plus TCS = 56 patients, tezepelumab plus TCS = 55 patients	Adults, 18 to 75 years of age	Pruritus Numerical Rating and 5-D itch scales	No statistically significant improvement. Peak pruritus NRS scores were numerically lower for tezepelumab plus TCS-treated patients at week 12 but did not reach nominal significance
Lebrikizumab	IL-13	Subcutaneous injection							
Simpson E.L. et al. *JAAD*, 2018 [100]			Phase 2 study (TREBLE)	1:1:1:1 to receive lebrikizumab 125 mg single dose at baseline, 250 mg single dose at baseline, 125 mg once every 4 weeks, or placebo every 4 weeks plus TCS for 12 weeks	12-week study	212 patients in total, lebrikizumab 125 mg single dose, *n* = 52, lebrikizumab 250 mg single dose, *n* = 53, lebrikizumab 125 mg Q4W, *n* = 51, placebo, *n* = 53	Adults, 18–75 years	Pruritus Visual Analog Scale (VAS) score	Improvement on pruritus in patients receiving lebrikizumab.
Guttman-Yassky E. et al. *JAMA Dermatol.* 2020 [101]			Phase 2b study	280 patients randomized to placebo (*n* = 52) or to lebrikizumab at doses of 125 mg every 4 weeks (*n* = 73), 250 mg every 4 weeks (*n* = 80), or 250 mg every 2 weeks (*n* = 75).	16-week study	280 patients placebo (*n* = 52), lebrikizumab at doses of 125 mg (*n* = 73), 250 mg every 4 weeks (*n* = 80), or 250 mg every 2 weeks (*n* = 75).	Adults 18 years or older	Pruritus Numerical Rating Scale (NRS) score	Lebrikizumab improved pruritus in a dose-dependent manner vs placebo during 16 weeks of treatment.
Dupilumab	IL-4Rα	Subcutaneous injection							
Blauvelt A. et al. *Lancet* 2017 [102]			Phase 3 trial	Patients were assigned randomly to 3:1:3. In total, 740 patients were enrolled; 319 received dupilumab qw plus topical corticosteroids, 106 received dupilumab q2w plus topical corticosteroids, and 315 received placebo plus topical corticosteroids.	16-week study	In total, 740 patients were enrolled; 319 received dupilumab qw plus topical corticosteroids, 106 received dupilumab q2w plus topical corticosteroids, and 315 received placebo plus topical corticosteroids.	Adult patients, aged 18 years old or older	NRS	Patients receiving dupilumab plus TCS had a greater improvement in peak NRS pruritus than those receiving placebo.
Tradipitant (VLY-686)	a novel Neurokinin (NK)-1 receptor antagonist	Oral systemic medication							
Welsh S.E. et al. *JEADV* 2021 [103]			A phase 3, randomized, placebo-controlled trial (EPIONE)		8-week study	375 patients, tradipitant (*n* = 188) or placebo (*n* = 187)		WI-NRS: Worst Itch Numeric Rating Scale	EPIONE did not meet its primary endpoint of reduction in pruritus. However, robust antipruritic effect was observed in patients with mild lesion severity. Tradipitant treatment resulted in a clinically meaningful reduction in patient-reported worst itch.

## Data Availability

No new data were created or analyzed in this study. Data sharing is not applicable to this article.

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
