# Peer review of "Pruritogenic Mediators and New Antipruritic Drugs in Atopic Dermatitis"

_jcm, 2023, doi:10.3390/jcm12062091_

Round 1

Reviewer 1 Report

1.The article is well written.

More data in introduction about AD in elderly people reccurence are of interest .Recent data should be added like DOI: 10.1016/j.abd.2022.01.011

and  DOI: 10.1016/j.abd.2022.01.011  DOI: 10.1093/bjd/ljac097

2.Recent data related to AD ,IGE level ,LDH level,triptase role/levels and pruritus are of interest and should be added DOI: 10.12659/MSM.937990

3.Data on JAK inhibitors,targets and pruritus should be added exDOI: 10.1007/s40257-022-00748-2  DOI: 10.3389/fmed.2022.1079323

4.New data on IL 6,4,13,31,33 cutaneous and systemic involvement related to pruritus and other organs influence are of interest and recent data should be added DOI: 10.3892/etm.2021.9693 DOI: 10.2147/JIR.S374060

5.Interrelations between cutaneous microbioma and pruritus directly or through TLR should be added too  DOI: 10.3390/cells11233930 DOI: 10.1016/j.brainresbull.2022.09.014

Author Response

Reply to reviewer

Dear reviewer,

Thank you for giving us the possibility to revise and improve our manuscript.

1.The article is well written.

More data in introduction about AD in elderly people reccurence are of interest .Recent data should be added like DOI: 10.1016/j.abd.2022.01.011

Now, more data regarding AD in elderly have been added.

The prevalence of AD among patients 60 to 69 years old in Saarland, Germany, was 4% [7]. In Poland, the prevalence of AD was 2% for those over 60 years old, compared to 5% for children (ages 6-7), 4% for teenagers (ages 13–14), and 3% for adults (20–44 years) [8]. Similarly, 4% of adults (30-39 years) and 3% of elderly patients (60 years or older) who attended a hospital at Tokyo had AD [9] compared to 19% of youngsters (7–9 years) at Otsu, Japan [10, 11].

and  DOI: 10.1016/j.abd.2022.01.011  DOI: 10.1093/bjd/ljac097

2.Recent data related to AD ,IGE level ,LDH level,triptase role/levels and pruritus are of interest and should be added DOI: 10.12659/MSM.937990

Also recent data in AD, IgE, LDH and tryptase have been added

Increased levels of both total serum IgE and specific IgE to common allergens are frequently present in these patients. Based on this hyperreactivity, two separate subtypes of AD have recently been identified: an "intrinsic" form of AD that lacks detectable IgE-mediated sensitization and a "extrinsic" form linked to IgE-mediated sensitization to environmental allergens [20]. According to recent studies, the prevalence of extrinsic and intrinsic AD is, respectively, 70–80% and 20–30%. Environmental allergens including animal dander, grasses, and pollens, as well as certain foods like milk, eggs, and wheat, can exacerbate AD [11]. Children are more susceptible to food allergy-related symptoms than older patients, while adult and elderly AD patients with moderate-to-severe disease tend to be more sensitive to food allergens than patients with milder AD [11]. In a study with AD patients, six inhaled allergens were most frequently found in patients with high allergen-specific IgE (44 or 73.3%), with the majority of allergens coming from timothy grass pollen (43.2%). The second allergen most frequently found (25.0%) was birch pollen. Extremely high levels of total IgE, thymus and activation-regulated chemokine levels, and peripheral blood eosinophil counts were present in elderly patients with AD [21]. Cross-reactions may be the cause of the high incidence of these inhaled allergens as both timothy grass and birch grass pollen have many elements in common with other allergens. Birch pollen frequently reacts adversely with allergens in fruits, nuts, and vegetables [22]. The levels of total IgE, but the tryptase levels, have been associated with the severity of the symptoms in atopic dermatitis patients. [23]. More effective candidate biomarkers for the severity of AD may be serum LDH and IL-18 [24]. Older AD patients are less allergic to foods, animal dander, and fungi, and more sensitive to dust mites and pollens, as indicated by elevated serum IgE levels specific to these allergens [25]. The relationship between serum IgE and treatment response has been controversial. Some studies have shown that dynamic changes in serum IgE levels may reflect the way AD responds to treatment, while others have shown that there is only a weak relationship between serum IgE and follow-up disease severity after treatment, suggesting that serum IgE is not the most accurate biomarker of AD [25].

3.Data on JAK inhibitors,targets and pruritus should be added ex DOI: 10.1007/s40257-022-00748-2  DOI: 10.3389/fmed.2022.1079323

Data on JAK inhibitors and pruritus have been added

3.11 Janus kinase ( JAK) /Signal Transducer and Activator of Transcription ( STAT)

The Janus kinases (JAKs) are protein tyrosine kinases (TYKs) that bind to transmembrane type 1 and type 2 cytokine receptors and mediate cellular responses to numerous cytokines and growth factors; these mediators are important in immune defense and in immune-mediated disease. [196].

Janus kinases (JAKs) are a family of kinases that include JAK1, JAK2, JAK3, and tyrosine kinase 2 (TYK2); JAKs mediate cytokine signaling by activating signal transducer and activator of transcription (STAT) transcription levels. The JAK/STAT pathway is activated in a variety of disease states. [197].

Cytokines including IL-31, IL-4, IL-13, and TSLP that are crucial for pruritus in AD transmit their signals into the cells via JAK-1 and JAK-2 [198]. In animal studies, JAK-1/2 inhibition or deletion greatly decreased the itch signals brought on by these mediators [198].  When the oral JAK 1/3 inhibitor tofacitinib greatly decreased pruritus in elderly patients with chronic pruritus of unknown cause, it was demonstrated that JAKs play a crucial role in human pruritus. In AD, inflammation, and pruritus are influenced by cytokines such as  IL-31, IL-4, IL-13, TSLP, and IL-5. A new therapy option is made possible by the possibility to inhibit JAK-1 and JAK-2 with specific JAK inhibitors. [198].

Abrocitinib, an oral selective JA-1 inhibitor, has been proven to reduce significantly pruritus in AD [199-202].  In a randomized, phase 2b, double-blinded, placebo-controlled trial in adult patients with AD, abrocitinib (200 mg, 100 mg, 30 mg, or 10 mg) or a placebo were given to participants at random and once daily for 12 weeks. In comparison to the placebo group, differences in pruritus were seen in the 200-mg and 100-mg groups [199]. In a further double-blind, randomised phase 3 trial (JADE MONO-1), with patients with moderate to severe AD and age more than 12 years, abrocitinib 200 mg, abrocitinib 100 mg, or a placebo was given to patients at random (2:2:1) once a day for 12 weeks. Between the start of the treatment and week 12, a considerable, quick (i.e., within 2 days) reduction in the severity of pruritus and other atopic dermatitis symptoms was also noticed [200]. In a phase 3 trial, in terms of itch response at week 2, abrocitinib 200 mg was superior to dupilumab, but not the 100 mg dose. For the majority of other important secondary end-point comparisons, neither abrocitinib dose at week 16 substantially varied from dupilumab [202].  

Baricitinib, an oral JAK inhibitor that reversibly inhibits JAK1 and JAK2 and moderately inhibits TYK2 has been shown to reduce pruritus in AD patients [203, 204].

Furthermore, upadacibitinib, an oral JAK-1 inhibitor, has effectively reduced pruritus in AD [205, 206].  With each upadacitinib dosage regimen of 7.5, 15, or 30 mg once daily, a significant improvement in pruritus from the baseline at week 16 was also observed compared to the placebo in a clinical trial [205].    

Topical delgocitinib is an investigational JAK inhibitor that blocks tyrosine kinase 2, JAK1, JAK2, and JAK 3. When given to Japanese adult patients with moderate to severe atopic dermatitis for up to 28 weeks, delgocitinib 0.5% ointment, a new topical Janus kinase inhibitor, improved clinical signs and symptoms with a positive safety profile [207].   Nakagawa H. Et al, 2020 JAAD. At week 1, the delgocitinib group had a lower pruritus numeric rating scale (NRS) than the vehicle group, and this difference persisted over time in adults and children [207, 208].    Tofacitinib reversibly inhibits JAK1 and JAK3 in vitro and to a lesser extent inhibits JAK2 and TYK2 [209].  Tofacitinib, by inhibition of JAK3, inhibits cytokines IL-2, IL-4, IL-7, IL-9, IL-15, and IL-21; and by JAK1 inhibition, inhibits IL-6, type 1 IFNs, and IFN-gamma. In a phase 2a, randomized trial, the effectiveness of topical tofacitinib for the treatment of atopic dermatitis was assessed. In this trial, tofacitinib 2% ointment or placebo was applied twice daily for a period of four weeks to 69 adult patients with clinically stable, mild to moderate atopic dermatitis. Significant improvements in pruritus in the tofacitinib group were observed by day 2 [210, 211].    

A topical version of the drug ruxolitinib, which is a strong and specific inhibitor of Janus kinase (JAK)1 and JAK2, is available as ruxolitinib cream 1.5% (OPZELURATM) that has been used topically in atopic dermatitis [212]. Ruxolitinib cream 1.5%, when used twice daily for eight weeks, significantly reduced indices of disease severity, pruritus, and sleep disturbance in two identically constructed, international, phase III studies in patients with mild to severe AD aged 12 years [212-214].    

4.New data on IL 6,4,13,31,33 cutaneous and systemic involvement related to pruritus and other organs influence are of interest and recent data should be added DOI: 10.3892/etm.2021.9693 DOI: 10.2147/JIR.S374060

Now these new data have been added

Niculet E, Chioncel V, Elisei AM, Miulescu M, Buzia OD, Nwabudike LC, Craescu M, Draganescu M, Bujoreanu F, Marinescu E, Arbune M, Radaschin DS, Bobeica C, Nechita A, Tatu AL. Multifactorial expression of IL-6 with update on COVID-19 and the therapeutic strategies of its blockade (Review).Exp Ther Med. 2021 Mar;21(3):263. doi: 10.3892/etm.2021.9693. Epub 2021 Jan 25.PMID: 33603870 

Tatu AL, Nadasdy T, Arbune A, Chioncel V, Bobeica C, Niculet E, Iancu AV, Dumitru C, Popa VT, Kluger N, Clatici VG, Vasile CI, Onisor C, Nechifor A. Interrelationship and Sequencing of Interleukins4, 13, 31, and 33 - An Integrated Systematic Review: Dermatological and Multidisciplinary Perspectives.J Inflamm Res. 2022 Sep 8;15:5163-5184. doi: 10.2147/JIR.S374060. eCollection 2022.PMID: 36110506 

  1. Interrelations between cutaneous microbiome and pruritus directly or through TLR should be added too DOI: 10.3390/cells11233930 DOI: 10.1016/j.brainresbull.2022.09.014

Also the Interrelations between cutaneous microbiome and pruritus directly or through TLR

Moniaga CS, Tominaga M, Takamori K. An Altered Skin and Gut Microbiota Are Involved in the Modulation of Itch in Atopic Dermatitis. Cells. 2022 Dec 5;11(23):3930. doi: 10.3390/cells11233930.PMID: 36497188

Zhang Q, Li T, Niu J, Xiao J, Zhang MZhang R, Chen D, Shi Y, Zhang X, Hu X, Yu B, Feng J, Fang Q. Inhibitory effects of antibiotic-induced gut microbiota depletion on acute itch behavior in mice.

Brain Res Bull. 2022 Nov;190:50-61. doi: 10.1016/j.brainresbull.2022.09.014. Epub 2022 Sep 17.PMID: 36126873

These references have now been added to the manuscript.

Thank you for your comments.

Mitsuyama S, Higuchi T. Effectiveness of dupilumab for chronic prurigo in elderly patients with atopic dermatitis. An Bras Dermatol. 2023 Jan-Feb;98(1):86-89. doi: 10.1016/j.abd.2022.01.011. Epub 2022 Nov 11.PMID: 36376117 

Yue W., Cheng D., Sun Z., Shen Y., Wang S., Liu X, Pei X., Deng S., Pan H., Liao Z., Li W., Yao X.,  Liang Y., Song Z.,  Yao Z.,  Zhang H., and Guo Y. Validation of diagnostic criteria for atopic dermatitis and proposal of novel diagnostic criteria for adult and elderly Chinese populations: a multicentre, prospective, clinical setting-based study. Br J Dermatol. 2022 Dec 12;ljac097. doi: 10.1093/bjd/ljac097. Online ahead of print.

Černiauskienė M, Bagdonaitė L, Karčiauskaitė D, Kvedarienė V. Prevalence and Variability of Allergen-Specific Immunoglobulin E in Patients with Elevated Tryptase Levels.

Med Sci Monit. 2022 Dec 14;28:e937990. doi: 10.12659/MSM.937990.PMID: 36514263 

Hoy SM. Ruxolitinib Cream 1.5%: A Review in Mild to Moderate Atopic Dermatitis.

Am J Clin Dermatol. 2023 Jan;24(1):143-151. doi: 10.1007/s40257-022-00748-2. Epub 2022 Dec 20.PMID: 36538235 

Rodriguez-Le Roy Y, Ficheux AS, Misery L, Brenaut E. Efficacy of topical and systemic treatments for atopic dermatitis on pruritus: A systematic literature review and meta-analysis.

Front Med (Lausanne). 2022 Dec 22;9:1079323. doi: 10.3389/fmed.2022.1079323. eCollection 2022.PMID: 36619624 

Reviewer 2 Report

This is such an important and interesting topic. However, the authors do not follow the updates in this field well. Some excellent work from Dr. Diana Bautista, Dr. Martin Steinhoff, and Dr. Brian Kim, etc. is not cited. Meanwhile, some language is not rigorous and a number of typos can be found. 

Author Response

This is such an important and interesting topic. However, the authors do not follow the updates in this field well. Some excellent work from Dr. Diana Bautista, Dr. Martin Steinhoff, and Dr. Brian Kim, etc. is not cited. Meanwhile, some language is not rigorous and a number of typos can be found. 

Reply to reviewer

Dear reviewer,

Thank you for giving us the possibility to revise and improve our manuscript.

These references have now been added to the manuscript.

Thank you for your comments.

Mali SS, Bautista DM. Basophils add fuel to the flame of eczema itch.

Cell. 2021 Jan 21;184(2):294-296. doi: 10.1016/j.cell.2020.12.035.PMID: 33482094 

Wilson SR, Thé L, Batia LM, Beattie K, Katibah GE, McClain SP, Pellegrino M, Estandian DM, Bautista DM. The epithelial cell-derived atopic dermatitis cytokine TSLP activates neurons to induce itch. Cell. 2013 Oct 10;155(2):285-95. doi: 10.1016/j.cell.2013.08.057. Epub 2013 Oct 3.PMID: 24094650 

Morita T, McClain SP, Batia LM, Pellegrino M, Wilson SR, Kienzler MA, Lyman K, Olsen AS, Wong JF, Stucky CL, Brem RB, Bautista DM. HTR7 Mediates Serotonergic Acute and Chronic Itch.

Neuron. 2015 Jul 1;87(1):124-38. doi: 10.1016/j.neuron.2015.05.044. Epub 2015 Jun 11.PMID: 26074006 

Wilson SR, Gerhold KA, Bifolck-Fisher A, Liu Q, Patel KN, Dong X, Bautista DM. TRPA1 is required for histamine-independent, Mas-related G protein-coupled receptor-mediated itch. Nat Neurosci. 2011 May;14(5):595-602. doi: 10.1038/nn.2789. Epub 2011 Apr 3.PMID: 21460831 

Steinhoff M, Ahmad F, Pandey A, Datsi A, AlHammadi A, Al-Khawaga S, Al-Malki A, Meng J, Alam M, Buddenkotte J. Neuroimmune communication regulating pruritus in atopic dermatitis. J Allergy Clin Immunol. 2022 Jun;149(6):1875-1898. doi: 10.1016/j.jaci.2022.03.010. Epub 2022 Mar 23.PMID: 35337846

Ständer S, Steinhoff M. Pathophysiology of pruritus in atopic dermatitis: an overview. Exp Dermatol. 2002 Feb;11(1):12-24. doi: 10.1034/j.1600-0625.2002.110102.x.PMID: 11952824 

Trier AM, Mack MR, Fredman A, Tamari M, Ver Heul AM, Zhao Y, Guo CJ, Avraham O, Ford ZK, Oetjen LK, Feng J, Dehner C, Coble D, Badic A, Joshita S, Kubo M, Gereau RW 4th, Alexander-Brett J, Cavalli V, Davidson S, Hu H, Liu Q, Kim BS.
IL-33 signaling in sensory neurons promotes dry skin itch.

J Allergy Clin Immunol. 2022 Apr;149(4):1473-1480.e6. doi: 10.1016/j.jaci.2021.09.014. Epub 2021 Sep 21.PMID: 34560104 

Kim BS. The translational revolution of itch.

Neuron. 2022 Jul 20;110(14):2209-2214. doi: 10.1016/j.neuron.2022.03.031. Epub 2022 Apr 20.PMID: 35447089 

Kim BS, Howell MD, Sun K, Papp K, Nasir A, Kuligowski ME; INCB 18424-206 Study Investigators. Treatment of atopic dermatitis with ruxolitinib cream (JAK1/JAK2 inhibitor) or triamcinolone cream.

J Allergy Clin Immunol. 2020 Feb;145(2):572-582. doi: 10.1016/j.jaci.2019.08.042. Epub 2019 Oct 17.PMID: 31629805 

Yang TB, Kim BS. Pruritus in allergy and immunology.

J Allergy Clin Immunol. 2019 Aug;144(2):353-360. doi: 10.1016/j.jaci.2019.06.016.PMID: 31395149 

Reviewer 3 Report

This review is well-organized.

I suggest that the author could add a table about the treatment targeted on these mediators (new antipruritic drugs), and a figure about the pathogenesis of these mediators. It would be more reader friendly and informative.

Author Response

Reviewer 3

This review is well-organized.

I suggest that the author could add a table about the treatment targeted on these mediators (new antipruritic drugs), and a figure about the pathogenesis of these mediators. It would be more reader-friendly and informative.

Reply to reviewer

Dear reviewer,

Thank you for giving us the possibility to revise and improve our manuscript.

Now, a table ( Table 2) has been added regarding these new antipruritic drugs ( mediators).

It was difficult to create a  figure of the pathogenesis of these mediators because of the copyright but I have created instead a Table ( Table 1).

Thank you for your comments.

Reviewer 4 Report

A well organized review article. A table showing mediators responsible for chronic pruruitus in AD and potential drugs may be added 

In the 2. paragraph,lines 37-39  seem needless, can be removed

Author Response

A well organized review article. A table showing mediators responsible for chronic pruruitus in AD and potential drugs may be added 

In the 2. paragraph,lines 37-39  seem needless, can be removed

Reply to reviewer

Dear reviewer,

Thank you for giving us the possibility to revise and improve our manuscript.

Now, a table has been added showing mediators responsible for chronic pruritus in AD ( Table 1).

Also, another table has been added regarding these new antipruritic drugs ( mediators), Table 2.

In paragraph 2, lines 37-39 have been removed.

Thank you for your comments

Round 2

Reviewer 2 Report

The article is improved. However, since it is a review, could authors read the publications more carefully? Please cite those original and scientific publications. For instance, why doe dupilumab excert a good anti-itch effect? Reference 83 is a pre-review paper.  Could authors use some figures to combine the disease mechanism and medication targets?

Author Response

Dear Reviewer,

Thank you for giving us the possibility to revise and improve our manuscript.

We have now performed the requested changes and also we have designed 2 new figures with mediators and medication targets of pruritus in atopic dermatitis.

The new changes are marked in blue.

We are looking forward to hearing from you

Yours sincerely,

Dimitra Koumaki

Reviewer’s comment

The article is improved. However, since it is a review, could authors read the publications more carefully? Please cite those original and scientific publications. For instance, why does dupilumab exert a good anti-itch effect? Reference 83 is a pre-review paper.  Could authors use some figures to combine the disease mechanism and medication targets?

Reply

Thank you for giving us the possibility to revise and improve our manuscript.

Now, the original publications have been added. In reference 83, the original article has been cited. Also, the original dupilumab trial LIBERTY AD CHRONOS has been cited ( reference 110). The new changes are marked in blue.

References:

  1. Wang F, Trier AM, Li F, Kim S, Chen Z, Chai JN, Mack MR, Morrison SA, Hamilton JD, Baek J, Yang TB, Ver Heul AM, Xu AZ, Xie Z, Dong X, Kubo M, Hu H, Hsieh CS, Dong X, Liu Q, Margolis DJ, Ardeleanu M, Miller MJ, Kim BS. A basophil-neuronal axis promotes itch. Cell. 2021 Jan 21;184(2):422-440.e17. doi: 10.1016/j.cell.2020.12.033. Epub 2021 Jan 14.PMID: 33450207
  2. Blauvelt A, de Bruin-Weller M, Gooderham M, Cather JC, Weisman J, Pariser D, Simpson EL, Papp KA, Hong HC, Rubel D, Foley P, Prens E, Griffiths CEM, Etoh T, Pinto PH, Pujol RM, Szepietowski JC, Ettler K, Kemény L, Zhu X, Akinlade B, Hultsch T, Mastey V, Gadkari A, Eckert L, Amin N, Graham NMH, Pirozzi G, Stahl N, Yancopoulos GD, Shumel B. Long-term management of moderate-to-severe atopic dermatitis with dupilumab and concomitant topical corticosteroids (LIBERTY AD CHRONOS): a 1-year, randomised, double-blinded, placebo-controlled, phase 3 trial. Lancet. 2017 Jun 10;389(10086):2287-2303. doi: 10.1016/S0140-6736(17)31191-1. Epub 2017 May 4. PMID: 28478972

We have also created two new figures regarding the pathogenesis, mediators, and target therapies of pruritus: Figure 1 and Figure 2.

Please let us know if anything is needed.

We are looking forward to hearing from you

Best wishes

Dimitra Koumaki